# YOLO-SS-Large: A Lightweight and High-Performance Model for Defect Detection in Substations

**DOI:** 10.3390/s23198080

**Published:** 2023-09-26

**Authors:** Qian Wang, Lixin Yang, Bin Zhou, Zhirong Luan, Jiawei Zhang

**Affiliations:** 1Qujiang Campus, School of Electrical Engineering, Xi’an University of Technolgy, Xi’an 710048, China; 2211921161@stu.xaut.edu.cn (L.Y.); luanzhirong@xaut.edu.cn (Z.L.); jiawei8633@163.com (J.Z.); 2North China Electric Power Research Institute Co., Ltd. Xi’an Branch, Xi’an 710000, China; zhoubin19752023@163.com

**Keywords:** substation defect detection, YOLOv5, FasterNet, NWD-CIoU, dynamic head

## Abstract

With the development of deep fusion intelligent control technology and the application of low-carbon energy, the number of renewable energy sources connected to the distribution grid has been increasing year by year, gradually replacing traditional distribution grids with active distribution grids. In addition, as an important component of the distribution grid, substations have a complex internal environment and numerous devices. The problems of untimely defect detection and slow response during intelligent inspections are particularly prominent, posing risks and challenges to the safe and stable operation of active distribution grids. To address these issues, this paper proposes a high-performance and lightweight substation defect detection model called YOLO-Substation-large (YOLO-SS-large) based on YOLOv5m. The model improves lightweight performance based upon the FasterNet network structure and obtains the F-YOLOv5m model. Furthermore, in order to enhance the detection performance of the model for small object defects in substations, the normalized Wasserstein distance (NWD) and complete intersection over union (CIoU) loss functions are weighted and fused to design a novel loss function called NWD-CIoU. Lastly, based on the improved model mentioned above, the dynamic head module is introduced to unify the scale-aware, spatial-aware, and task-aware attention of the object detection heads of the model. Compared to the YOLOv5m model, the YOLO-SS-Large model achieves an average precision improvement of 0.3%, FPS enhancement of 43.5%, and parameter reduction of 41.0%. This improved model demonstrates significantly enhanced comprehensive performance, better meeting the requirements of the speed and precision for substation defect detection, and plays an important role in promoting the informatization and intelligent construction of active distribution grids.

## 1. Introduction

With the expansion of the power grid, the number of devices in substations has been increasing, making the detection of internal defects in substations increasingly important. The intelligent identification of substation defects is essentially an object detection problem, which aims to determine the position and type of specific objects in images. This topic has a long research history in the academic community [1] (Wang X, 2017). Traditional two-stage detection methods are based on the RCNNs (region-based convolutional neural network) framework, which divides the object detection task into two stages: region proposal extraction and object classification with localization. In the region proposal extraction stage, these methods use algorithms to generate a series of candidate regions that may contain objects [2]. Common methods for generating candidate regions include Selective Search and EdgeBoxes. These algorithms generate candidate regions based on low-level image features such as color, texture, edges, and contextual information. In the object classification and localization stage, each candidate region is fed into a deep learning model for object classification and position regression. The most well-known two-stage detection methods are based on the RCNN framework, including RCNN, Fast RCNN, Faster RCNN, etc. In the RCNN method, each candidate region is first adjusted to a fixed size and then features are extracted using convolutional neural networks. These features are then input to a fully connected layer for object classification and position regression. During training, the loss function is calculated and the model parameters are optimized using positive and negative samples.

Traditional RCNN adopts a two-stage network structure, which achieves high precision and good stability in object detection [1]. However, it is difficult to avoid the drawback of slow detection speed. To address this issue, scholars have proposed SPPNet [2] and Fast-RCNN [3], which can effectively reduce the workload of feature extraction and improve detection speed [4,5,6]. In 2016, Joseph Redmon et al. proposed a single-stage network structure for object detection called “You Only Look Once (YOLO)” [7]. With the updates of YOLO versions, both its precision and computational speed have been continuously improved, making it a widely recognized object detection algorithm.

On the one hand, the rich redundant features in the YOLO-series algorithms ensure a comprehensive interpretation of the input by the model. These redundant features obtained through convolutional calculations not only guarantee the generalization ability of the model but also increase the model’s complexity. Therefore, many researchers have conducted research on lightweight methods in recent years. Zhang W et al. combined depth-separable convolution with point convolution and added batch normalization layers [8], which accelerated model convergence. H Xu et al. weighted each channel using the coordinate attention (CA) mechanism [9] to remove redundant features and effectively reduce the number of parameters. Zhou X et al. proposed a DWSC-YOLO model that added heterogeneous convolutions and improved activation functions [10]. This model had the same mAP value as YOLOv5 but reduced its volume by 79.8%. W Ji et al. used the lightweight network Shufflenetv2 with the convolutional block attention module (CBAM) as the backbone and added the adaptive spatial feature fusion (ASFF) module in the PANet network to improve precision [11]. There are numerous novel neural network architectures for lightweight YOLO algorithms, such as MobileNetV3 with the SE (squeeze excitation) attention mechanism and hand switch function, PP-LCNet with a similar structure, GhostNet, etc. [12,13,14,15,16,17]. B Ma et al. proposed a Gaussian distance intersection over union (GDIoU) loss function and applied it to the YOLOV4 network [18], which increased the average precision by 7.37%. Liu D et al. designed a new bounding box regression loss function that incorporates distance penalty for union and angle penalty for intersection [19]. They improved the YOLOV4-CSP network by combining this loss function with an adaptive angle setting method based on the K-means clustering algorithm, significantly improving mAP and FPS compared to traditional networks. These improvements, to some extent, solve the problem of the degradation of the loss function to the intersection over union metric under certain specific position relationships between predicted bounding boxes and ground truth bounding boxes. However, they overlook the uneven contribution of training samples to the loss, poor regression effects, and slow convergence speed of the loss function.

On the other hand, at the current stage, almost all advanced object detectors adopt the same pattern: a backbone network for feature extraction and detection heads for localization and classification tasks [20,21,22]. Song L et al. proposed a delicate dynamic head that selects features from different scales using a feature pyramid network and extracts pixel-level combinations [23], further enhancing the representation ability of multi-scale features. Goindani A et al. improved the importance of each detection head by adding additional attention layers and utilizing the input and output of multi-head attention [24]. This method outperforms traditional approaches, especially when training data are limited. These methods to some extent enhance the representation ability of the detection heads, but how to effectively integrate the awareness ability of multiple scales remains an unresolved issue.

In summary, to address the issues of the uneven contribution of training samples and the poor integration effect of awareness ability in detection heads, this paper proposes a novel fusion loss function called NWD-CIoU. It combines the lightweight structure of FasterNet and the dynamic head module to build a high-performance lightweight improved model, YOLO-Substation-large (YOLO-SS-Large), based on YOLOv5m. This model achieves an average precision of 67.6%, which is 0.3% higher than that of YOLOv5m. It achieves an FPS of 196.86, which is 43.5% higher than that of YOLOv5m. The parameter number is reduced to 12.36M, which is 41.0% lower than that of YOLOv5m.

## 2. Materials and Methods

In this chapter, a dataset of substation images is constructed, and the distribution of classes in the dataset is described in detail. Additionally, this paper introduces the network structure and loss function of the basic YOLOv5 model. It analyzes the drawbacks of the model in terms of parameter number, loss function, and detection heads and explains the motivation for improvements.

### 2.1. Database for This Paper

The equipment defects in substations mainly include damaged meters, transformer oil leakage, damaged breathers, cracked insulators, floating suspended objects, and the abnormal closure of cabinet doors. During the regular inspection of the equipment inside substations, any abnormal operating conditions that are not promptly detected or equipment defects that are not timely discovered can potentially lead to equipment failures and even different levels of power accidents. For example, the main function of breathers is to filter moisture from the air, ensuring that the air inside the transformer oil pillow remains dry and clean. If the breathers fail due to saturated moisture adsorption and are not replaced in a timely manner, this can easily cause the transformer oil to become damp and oxidized, resulting in a decrease in insulation strength and internal transformer failures. The failure of insulators can cause protective actions to trip, severely affecting the safe operation of the system. Additionally, there are hidden dangers such as blurry dials, damaged casings leading to abnormal readings, foreign objects floating and hanging on electrical wires causing three-phase short circuits, and ground discharges. To ensure the high-quality development of a new power system, no abnormal equipment defects within the substations that may affect the safe operation of the system can be ignored.

To achieve the intelligent identification of substation defects, a large number of relevant images need to be collected. Considering the various types of equipment defects in actual substations and the numerous unsafe behaviors of inspection personnel, this study focuses on a specific 220 kV substation. We further consider the serious hazards caused by numerous pieces of equipment in actual substations and the unsafe behavior of inspection personnel. We collected image data of unsafe behavior and safety hazards influenced by external factors in substations. With methods such as on-site photography (5634 images), capturing surveillance footage (2049 images), and collecting historical data (2894 images), a total of 10,577 images were collected. Most of these images contains a single-defect object, while a small portion contain multiple-detection objects in the same situation. There are 17 categories of defect detection. Table 1 shows the corresponding label IDs and numbers for each defect category.

In this paper, the LabelImg tool was used to create XML format labels, and these XML label files were then converted into the TXT format that can be recognized by the YOLO algorithm. Finally, the dataset was divided into training, testing, and validation sets in a ratio of 7:2:1. Figure 1 shows some examples of substation image data. Most of the images in the dataset contain multiple smaller-sized detection categories, which not only provide rich category features but also present a more realistic representation of the complex equipment environment in substations.

The distribution of classes in the training set of the substation image data is shown in Figure 2a; Figure 2b displays the proportions of training, testing, and validation sets for each category in the dataset. This visualization allows us to observe that the dataset division process satisfies the criteria of independent and identically distributed (i.i.d.) data. In other words, the distribution of data for each category across the three sets is uniform and independent. This finding further validates the reliability and authenticity of the training, testing, and validation processes of the model in this substation image dataset.

As shown in Figure 3a,b, the YOLOv5 model’s object detection head outputs the predicted offset of bounding box positions. In this context, x and y represent the coordinates of the predicted bounding box’s center, while width and height correspond to the predicted bounding box’s width and height, respectively. The darker the color in Figure 3, the more data is distributed in that area. The majority of the predicted boxes are distributed near the center of the image, with a prevalence of medium- and small-sized objects being detected.

### 2.2. YOLOv5 (3 Methods) and Disadvantage Analysis

On the one hand, the dataset of substation defects collected in this study contains a higher number of smaller-sized targets. Due to the complex environment of actual substations, it is difficult to avoid the problem of indistinguishable backgrounds in the images. YOLO series algorithms, benefiting from the pattern of generating candidate regions, greatly reduce the mistakes made by mistaking a background as an object target compared to R-CNN series algorithms. On the other hand, actual substations have limited computing resources and a high requirement for detection speed. The traditional R-CNN algorithm uses selective search to generate candidate regions, while the YOLO series algorithm replaces this part with a grid-based approach, directly mapping from the grid to regions. This reduces redundant computations and saves a significant amount of search time, resulting in faster detection speed. Therefore, the YOLO series algorithm demonstrates better engineering significance when applied to substation defect detection.

YOLOv5 is a deep learning-based object detection model that is an improvement over YOLOv4. The overall structure of the YOLOv5 network model consists of three parts: the backbone network for extracting multi-level feature maps, the neck network for collecting and merging different layer feature maps, and the object detection head for predicting object categories and positions.

Compared to YOLOv4, YOLOv5 has a faster detection speed under the same conditions, enabling higher real-time performance. YOLOv5 achieves significant improvements in precision, performs well on the COCO public dataset, and also demonstrates better robustness. YOLOv5 can be freely configured for different application situations, including different network structures, data augmentation methods, and training strategies.

The YOLOv5 network’s structure is shown in Figure 4. It utilizes the CSPDarknet53 structure as the backbone network, which includes convolutional modules CBS (Conv-BN-SiLU) and C3 modules, and the spatial pyramid pooling fast (SPPF) module. The C3 module consists of three standard convolutional layers and multiple bottleneck modules that reduce the number of channels in feature maps. The SPPF module performs dimensionality reduction through three different sized max pooling operations and then concatenates the channel numbers. The neck network adopts the structure design of the path aggregation network (PAN), including standard convolution modules, upsampling modules, and C3 modules without shortcut connections (shortcut = false).

Object detection algorithms rely on the backbone network to extract feature information, and YOLOv5 adopts the CSPDarknet structure as its backbone network. The rich redundant features ensure that the model fully interprets the input and improves the model’s precision. However, this network structure is complex and has a large number of parameters, which requires high hardware storage and computational power. Additionally, the redundant features obtained through convolutional computations consume a significant amount of computing resources while ensuring the model’s generalization ability. Therefore, lightweighting the backbone network is a necessary problem to address.

The total loss function of the YOLOv5 network consists of three components: localization loss $L_{\mathrm{box}}$; confidence loss $L_{\mathrm{obj}}$; and classification loss$L_{\mathrm{cls}}$. The mathematical expressions for these losses are shown in Equations (1)–(4).
(1)$$L_{\mathrm{total}}=L_{\mathrm{box}}+L_{\mathrm{obj}}+L_{\mathrm{ds}}$$
(2)$$L_{\mathrm{box}}=\lambda_{\mathrm{IoU}}\sum_{i=0}^{S^{2}}\sum_{j=0}^{B}l_{\mathrm{ij}}^{\mathrm{obj}}L_{\mathrm{CIoU}}$$
(3)$$L_{\mathrm{obj}}=\lambda_{\mathrm{cls}}\sum_{i=0}^{S^{2}}\sum_{j=0}^{B}{l_{\mathrm{ij}}^{\mathrm{obj}}\lambda_{c}(C_{i}-\hat{C_{i}})}^{2}+\lambda_{\mathrm{cls}}\sum_{i=0}^{S^{2}}\sum_{j=0}^{B}{l_{\mathrm{ij}}^{\mathrm{noobj}}\lambda_{c}(C_{i}-\hat{C_{i}})}^{2}$$
(4)$$L_{\mathrm{cls}}=-\sum_{i=0}^{S^{2}}\sum_{j=0}^{B}l_{\mathrm{ij}}^{\;\mathrm{obj}}\sum_{c\epsilon \mathrm{classes}}\lambda_{c}$$
where λ represents the overall loss balance coefficient, l represents the weight of the loss for each class of samples, and $C_{i}$ represents the confidence score.

The computation of $L_{\mathrm{box}}$ involves three implementations: generalized intersection over union (GIoU), Distance-IoU (DIoU), and Complete-IoU (CIoU). In Figure 5, the region bounded by the yellow line represents the ground truth bounding box B, while the region enclosed by the black line represents the predicted bounding box $B_{\mathrm{gt}}$ by the model, The region enclosed in red represents the bounding box for calculating Euclidean geometric distance. The black dashed line in Figure 5 represents the Euclidean distance between the center points of the ground truth bounding box B and the predicted bounding box $B_{\mathrm{gt}}$, while the black solid line represents the diagonal distance between the ground truth bounding box B and the predicted bounding box $B_{\mathrm{gt}}$. The expression for the complete intersection over union (CIoU) loss function is shown in Equations (5) and (6).
(5)$$L_{\mathrm{CIoU}}=1-\mathrm{IoU}+\frac{\rho^{2}\left(B,B_{\mathrm{gt}}\right)}{c^{2}}+\alpha V$$
(6)$$V=\frac{4}{\pi^{2}}{(\arctan\frac{w_{\mathrm{gt}}}{h_{\mathrm{gt}}}-\arctan\frac{w}{h})}^{2}$$
where ρ represents the Euclidean distance between the predicted bounding box $B_{\mathrm{gt}}$ and the ground truth bounding box B, while c represents the diagonal distance of the minimum enclosing area. Additionally, α is the weight function that determines the importance of each component in the loss function. Meanwhile, V represents the aspect ratio of the detection box, which reflects the scale of its width and height.

The complete intersection over union (CIoU) loss function improves upon the distance intersection over union by introducing an additional penalty term that accounts for the aspect ratio of the predicted bounding box. This penalty term encourages the predicted bounding box to align more closely with the ground truth bounding box. 

However, the CIoU loss function overlooks two important aspects. First, the aspect ratio describes a relative value and can be somewhat ambiguous. Second, it does not consider the issue of imbalanced contributions of low-quality samples and high-quality samples to the loss. Therefore, it is particularly important to improve the loss function. It is crucial to address these limitations and improve the loss function.

In the YOLOv5 model, the object detection head module is primarily responsible for performing multi-scale object detection on the feature maps extracted by the backbone network. This module consists of three main parts: (1) anchors: anchors are used to define bounding boxes of different sizes and aspect ratios. They are typically obtained by clustering the ground truth bounding boxes in the training set using K-means clustering. The anchor values can be pre-computed and stored in the model for generating detection bounding boxes during inference. (2) Classification: this part is responsible for classifying each detection bounding box to determine whether it contains an object. It often uses a combination of fully connected layers and the Softmax activation function to classify the features. (3) Regression: the regression component is used to predict the position and size of each detection bounding box. It typically employs fully connected layers to perform regression on the features. However, this detection head often only achieves one type of awareness ability, such as scale awareness, spatial awareness, or task awareness. Therefore, further improvements are needed to simultaneously integrate all awareness abilities into the detection head.

## 3. Improved Model: YOLO-SS-Large

This chapter addresses three main limitations of the YOLOv5 basic model. Firstly, the C3 module in the original model is improved by adopting the lightweight design ideas from FasterNet to create a new network structure. This aims to reduce the model parameter number and computational complexity while enhancing its operation efficiency. Secondly, a new loss function called NWD-CIoU (Normalized Wasserstein Distance-CIoU) is introduced based on the lightweight model. This loss function considers the distance relationships and size information among bounding boxes, resulting in improved precision for object detection and better performance in terms of object localization and bounding box regression. Lastly, the Dyhead module is introduced to unify the scale-aware, spatial-aware, and task-aware attention of the object detection head. This enhances the model's ability to detect objects of different scales and improves its generalization ability in complex situations. With these improvements, the YOLO-SS-Large model is proposed to further enhance the comprehensive performance of object detection algorithms in terms of precision, speed, and effectiveness, addressing the limitations of the YOLOv5 basic model.

The complex C3 module in the YOLOv5 basic model increases the parameter number and computational complexity. By applying the lightweight design principles of FasterNet to improve the C3 module, the parameter number and computational complexity can be reduced, leading to faster computation and real-time performance. This is crucial for tasks that require fast object detection and deployment on resource-constrained devices.

The loss function used in the YOLOv5 basic model may not adequately consider the distance relationships and size information among bounding boxes, which can affect the precision of object localization and bounding box regression. The introduction of the NWD-CIoU loss function allows for better measurement of the precision of object detection results, thereby improving precision and localization ability. This is particularly important for applications that demand high-precision object detection, such as autonomous driving and surveillance systems. The object detection head in the YOLOv5 basic model may lack scale-awareness, spatial-awareness, and task-awareness attention, resulting in limited detection ability for objects of different scales and poor generalization in complex situations. The Dyhead module is introduced to unify the awareness ability of the object detection head, thereby enhancing the detection ability for objects of different scales and improving the model's adaptability and robustness in complex situations. This is crucial for improving the model’s adaptivity and robustness.

In conclusion, the reasons for addressing the three main limitations of the YOLOv5 basic model are to improve computational speed and real-time performance, enhance precision and localization ability, and strengthen adaptability and robustness. These improvements have great significance, allowing object detection algorithms to perform better in a wider range of applications and meet the diverse demands for object detection in various situations.

### 3.1. Light Weight Based on FasterNet

The number of the model’s parameters directly affects the computational resources required during the inference process. A practical model that aims to be applied should not only consider its outstanding performance in accuracy but also take into account its dependency on computational resources. However, the computational resources in actual substations are often limited, making it challenging to avoid the negative impact of large model sizes on real-time detection and embedded applications. To address these issues, this section draws inspiration from the lightweight design principles of FasterNet and attempts to use the C3-Faster module to construct a new YOLOv5m network architecture, aiming to achieve an effective reduction in model parameters.

FasterNet combines partial convolution (PConv) and point-by-point convolution, while reducing redundancy calculation, which refers to the redundant information added during data transmission or storage in order to improve data reliability. This redundancy information can be used to detect and repair errors in the data to improve the fault tolerance of the system and memory access, thus constructing lightweight networks with stronger and more effective spatial feature extraction. The working principles of PConv are illustrated in Figure 6: a conventional Conv is applied to a portion of the input channel for spatial feature extraction and leaves the rest of the channel unchanged. For continuous or distributed more conventional memory access, the first or last continuous channel is extracted as representative of the entire feature map for computation. The input and output feature maps are considered to have the same number of channels without loss of generality. As a result, the memory access for PConv is smaller. Figure 6 illustrates the overall FasterNet architecture. It contains four hierarchical levels with an embedding layer (4 × 4 convolution with a step size of 4) or a merging layer (regular 2 × 2 convolution with a step size of 2), whose function is to perform spatial downsampling and channel number expansion, and each stage of the network architecture has a FasterNet block. The last two stages of the FasterNet block have smaller memory access, and for this reason, more FasterNet blocks are placed. That is why the last two stages require more computation compared to the other stages. Each FasterNet block has 2 point-by-point convolutions (1 × 1 convolutions) after the Pconv layer. Together these blocks behave as inverted residual blocks, where the middle layer has an extended number of channels and prevents straightforward paths (shortcut) to reuse certain input features.

To cater to diverse computational resources and application situations, FasterNet offers four variant models: tiny, small, medium, and big. These variants are denoted as FasterNetT0/1/2, FasterNet-S, FasterNet-M, and FasterNet-L, respectively. While they share similar architectural principles, there are differences in terms of depth and width. The smaller variant models tend to utilize the GeLU activation function, while the larger ones employ the ReLU activation function. This choice is made based on the model size and complexity. Table 2 presents a comparison of the performance results for each version of the FasterNet models. It provides insight into the varying ability and efficiency of different model sizes.

The C3 module in the YOLOv5 backbone network consists of three standard convolutional layers and multiple bottleneck modules. After considering the trade-off between parameter number and precision, this paper adopts the lightweight concept of the FasterNet-T0 model. It replaces the Conv layers in the three CBS (Conv-BatchNorm-Activation) blocks of the C3 module with PConv (partial convolution) and PWConv (pointwise convolution) layers. This leads to the creation of the C3-Faster structure specifically designed for YOLOv5m, as shown in Figure 7.

Based on the C3-Faster model, this paper further proposes the FasterNet-YOLOv5m (F-YOLOv5m) model. The modified network structure of the F-YOLOv5m model is shown in Figure 8.

This structure combines the C3 module from the YOLO algorithm with PConv and PWConv operations. This structure applies conventional Conv for spatial feature extraction on the feature maps of some channels. Simultaneously, the PWConv operation is performed while keeping the remaining channels unchanged. This approach allows for the efficient utilization of information from all channels while significantly reducing memory access during training.

### 3.2. Improved Loss Function

Small object detection is common in real-world situations. However, traditional object detectors are primarily developed and researched for objects of regular sizes. The sensitivity of the intersection over union (IoU) metric varies greatly for objects of different scales. Fluctuations in IoU caused by minor positional changes can differ significantly for objects of various sizes. The discrete nature of IoU measurements lacks scale invariance, particularly for small objects. This phenomenon indicates two challenges. Firstly, the sensitivity of IoU on small objects leads to similar features between positive and negative samples, making it difficult for the network to converge. Secondly, since the IoU between predicted bounding boxes and arbitrary anchor boxes is below the threshold, each predicted bounding box is allocated less than one positive sample on average during training. This scarcity of supervision information makes it difficult to find an appropriate threshold for obtaining high-quality positive and negative samples.

To address these issues, this section improves the localization loss function based on F-YOLOV5m and proposes a novel fusion loss function called NWD-CIoU. This new loss function aims to alleviate the difficulties in training due to the sensitivity of IoU on small objects and the imbalance between positive and negative samples.

Normalized Wasserstein distance (NWD) is used to measure the similarity between predicted bounding boxes and ground truth bounding boxes. The calculation steps of NWD are as follows: first, the predicted bounding boxes and ground truth bounding boxes are modeled as two-dimensional Gaussian distributions. Then, NWD is employed to measure the similarity between these two distributions.

Compared to the traditional IoU loss function, NWD has several advantages. Firstly, NWD enables efficient and accurate similarity measurement between predicted bounding boxes and ground truth bounding boxes, regardless of whether there is stacking between small objects. Secondly, NWD is insensitive to objects of different scales, making it more suitable for measuring the similarity between small objects.

In summary, NWD offers an effective and precise way to measure the similarity between predicted and ground truth bounding boxes by modeling them as Gaussian distributions. It overcomes the limitations of traditional IoU loss function, allowing for efficient similarity measurement even when small objects do not exhibit a stack. Additionally, the scale invariance of NWD makes it better suited for measuring the similarity between small objects.

Real objects are unlikely to have perfect rectangular shapes. Therefore, for small objects, the bounding boxes may inevitably include some background pixels. Within a bounding box, foreground pixels typically concentrate in the center, while background pixels tend to concentrate on the edges. To better weight each pixel within the bounding box, the bounding box $R=(c_{x},c_{y},w,h)$ can be fitted into a two-dimensional Gaussian distribution N(μ,∑), as shown in Equation (7):(7)$$\mu =\left[\begin{array}{c} c_{x}\\ c_{y} \end{array}\right],\sum =\left[\begin{array}{cc} \frac{w^{2}}{4} & 0\\ 0 & \frac{h^{2}}{4} \end{array}\right]$$
where $\left(c_{x},c_{y}\right)$ represents the center coordinates of the bounding box and w and h represent the width and height of the bounding box, respectively.

By employing the Wasserstein distance from optimal transport theory, the distance between two distributions can be calculated. For two-dimensional Gaussian distribution, the expression for the second-order Wasserstein distance is shown in Equation (8):(8)$${W_{2}}^{2}(\mu_{1},\mu_{1})=m_{1}-\;{m_{22}}^{2}+Tr\left[\sum_{1}+\sum_{2}-2{\left(\sum_{2}^{1/2}\sum_{1}\sum_{2}^{1/2}\right)}^{1/2}\right]$$

By simplifying the Gaussian distributions $N_{g}$ and $N_{t}$, where $N_{g}$ represents the predicted bounding box $B_{g}=(c_{\mathrm{xg}},c_{\mathrm{yg}},w_{g},h_{g})$ and $N_{t}$ represents the ground truth bounding box $B_{t}=(c_{\mathrm{xt}},c_{\mathrm{yt}},w_{t},h_{t})$, the second-order Wasserstein distance between the two bounding boxes can be simplified as Equation (9):(9)$${W_{2}}^{2}(N_{g},N_{t})={\left({\left[c_{\mathrm{xg}},c_{\mathrm{yg}},\frac{w_{g}}{2},\frac{h_{g}}{2}\right]}^{T},{\left[c_{\mathrm{xt}},c_{\mathrm{yt}},\frac{w_{t}}{2},\frac{h_{t}}{2}\right]}^{T}\right)}_{2}^{2}$$
where ${W_{2}}^{2}(N_{g},N_{t})$ is a distance unit, while the threshold that represents the similarity among bounding boxes should be a proportion in the range of (0, 1). It is necessary to normalize ${W_{2}}^{2}(N_{g},N_{t})$ to obtain the *NWD*. The expression for *NWD* is shown in Equation (10). Furthermore, the calculation expression for the loss function based on *NWD* is presented in Equation (11):(10)NWD(Ng,Nt)=eW22(Ng,Nt)C
(11)$$L_{NWD}=1-NWD(N_{g},N_{t})$$

In the YOLOv5m model, the default choice for calculating the localization loss $L_{\mathrm{box}}$ is CIoU, which exhibits minimal regression errors in the majority of situations. However, considering that the dataset constructed in this paper does not exclusively consist of small objects, $L_{\mathrm{nwd}}$ was not directly used to replace $L_{\mathrm{CIoU}}$. Instead, by assigning an appropriate fusion weight r to $L_{\mathrm{nwd}}$ and $L_{\mathrm{CIoU}}$, NWD-CIoU loss function is proposed as a measurement criterion. The calculation expression for NWD-CIoU is shown in Equation (12):(12)$$L_{\mathrm{NWD}-\mathrm{CIoU}}=r\cdot L_{\mathrm{CIoU}}+(1-r)\cdot L_{\mathrm{NWD}}$$

### 3.3. Improvement of Detection Head

A high-performance object detection head should possess the following three abilities. Firstly, in the same image, multiple objects of different scales may exist simultaneously. Therefore, the detection head should have good scale-awareness. Secondly, in the same image, objects can appear with different shapes, rotations, and positions. Hence, the detection head should have good spatial awareness. Thirdly, different objects may require different task-specific representations, where different channels of feature maps correspond to different detection tasks. Thus, the detection head should have good task awareness. Building upon the previous sections, this subsection introduces a dynamic head module that combines three attention mechanisms of scale-awareness, spatial awareness and task awareness. This module is incorporated into the improved model through multiple nested layers.

In the dynamic head module, the three feature maps output by the neck network are treated as a three-dimensional tensor $F\epsilon R^{L\times S\times C}$. The general expression of self-attention is applied, as shown in Equation (13):(13)W(F)=π(F)·F
where π(·) is an attention function.

Based on the above Equation, tensor F undergoes learning process in all three dimensions: the difference in object scales is related to features at different levels, The cross-scale learning of tensor F facilitates scale awareness in object detection; various geometric transformations of different object shapes are related to features at different spatial positions, and learning of tensor F at different spatial positions benefits spatial awareness in object detection; different tasks can be associated with features in different channels, and cross-channel learning of tensor F is beneficial for task-awareness in object detection.

Implementing this attention mechanism through fully connected layers is a good solution. However, due to the high dimensionality of the tensor, directly learning an attention function for all dimensions would cause a significant computational cost, which is not feasible in practical engineering application. Therefore, the attention function is transformed into three consecutive attentions, with each attention focusing on one dimension only, as shown in Equation (14).
(14)$$W(F)=\pi_{c}(\pi_{s}(\pi_{l}(F)\cdot F)\cdot F)\cdot F$$
where $\pi_{c}\left(\cdot \right)$, $\pi_{s}\left(\cdot \right)$, and $\pi_{l}\left(\cdot \right)$ represent three different attention functions applied to scale, space, and task dimensions, respectively.

On one hand, objects with significant scale differences often co-exist in natural images, and detection objects of different scales correspond to feature maps of different sizes. However, features at different levels are usually extracted from different depths of the network, leading to noticeable semantic gaps. Therefore, changing the expressive power of different feature maps can enhance the scale-awareness ability of the model. This paper introduces a scale-aware attention module based on semantic importance to dynamically fuse features of different scales, allowing the importance of various feature levels to adapt to the input. The expression of this module is shown in Equation (15):(15)$$\pi_{l}(F)\cdot F=\sigma \left(f\left(\frac{1}{\mathrm{sc}}\sum_{s,c}F\right)\right)\cdot F$$
where f(x) represents a linear function that acts similar to a (1×1) convolutional layer; σ(x) represents the hard Sigmoid function, which is expressed as Equation (16):(16)$$\sigma (x)=\max\left[0,\min\left(1,\frac{x+1}{2}\right)\right]$$

On the other hand, detection objects can appear at arbitrary positions in an image, corresponding to feature maps of different spatial locations. In order to focus on the discriminative ability of different spatial positions, this paper introduces a spatial-awareness attention module based on fused features to focus on discriminative regions that co-exist consistently between spatial positions and feature levels. Considering the high dimensionality of space, this module is decomposed into two steps: first, using deformable convolution to sparsify the attention learning and then aggregating features across different levels at the same spatial position. This approach applies attention to each spatial position and adaptively aggregates multiple feature levels together to learn more discriminative representations. 

The process can be shown in Equation (17):(17)$$\pi_{s}(F)\cdot F=\frac{1}{L}\sum_{l=1}^{L}\sum_{k=1}^{K}w_{l,k}\cdot F(l,p_{k}+\Delta p_{k},c)\cdot \Delta m_{k}$$
where k represents the number of sparse sampling positions and $p_{k}+\Delta p_{k}$ denotes the position transferred by the self-learned spatial offset $\Delta p_{k}$, focusing on a discriminative region. $\Delta m_{k}$ is an important self-learned scalarat position $p_{k}$. Both $\Delta p_{k}$ and $\Delta m_{k}$ are learned from the input features at the median level of F.

In addition, detection objects can have different task-specific representations, often corresponding to different detection heads. To facilitate joint learning and ensure the generalization of object representations, a task-aware attention module is proposed: this module can dynamically open or close functional channels to support different tasks. This process can be represented as Equation (18). Firstly, global average pooling is performed on dimensions L×S to reduce dimensionality. Then, two fully connected layers and a normalization layer are used. Finally, a Sigmoid function is applied to normalize the output.
(18)$$\pi_{s}(F)\cdot F=\max(\alpha^{1}(F)\cdot F_{C}+\beta^{1}(F),\alpha^{2}(F)\cdot F_{C}+\beta^{2}(F))$$
where $F_{C}$ is the feature segment of the *c*-th channel, and $\theta \left(i\right)=\left[\alpha^{1},\alpha^{2},\beta^{1},\beta^{2}\right]$ is a hyper-function used to learn the threshold that controls the activation attention.

The above three attention mechanism modules are sequentially applied and can be nested multiple times. The detailed structure of the dynamic head module is shown in Figure 9. Firstly, the scale-aware attention module and spatial-aware attention module are applied to the feature pyramid. Then, the output is passed through an ROI pooling layer and replaces the original fully connected layer with the task-aware attention module. Finally, multiple dynamic head modules are stacked together.

In this paper, we utilize the dynamic head module to unify the attention mechanisms and construct the Dyhead-YOLOv5m architecture for YOLOv5, as shown in Figure 10.

This architecture adjusts the (80, 80, 256), (40, 40, 512), and (20, 20, 1024) feature layers extracted from the YOLOv5m neck network to the same scale, forming a three-dimensional tensor $F\epsilon R^{L\times S\times C}$ as the input to the dynamic head. Then, multiple dynamic head modules are stacked, including scale-aware, spatial-aware, and task-aware attention modules. The scale-aware module makes the feature maps more sensitive to the scale differences of foreground objects. The spatial-aware module sparsifies the feature maps and focuses on the discriminative spatial positions of foreground objects. The task-aware module reconfigures the feature maps into different activation features according to the requirements of downstream tasks. Finally, the output feature maps after multiple dynamic head modules are fed into the object detection head of the model to complete the object detection task.

## 4. Results and Discussion

### 4.1. Comparative Experiment of Lightweight

In this paper, under the condition of the same input feature map size, the YOLOv5m model is used as the basic network. The training performance of F-YOLOv5m was compared with three typical lightweight structures: MobileNetV3-YOLOv5m (M-YOLOv5m), PP-LCNet-YOLOv5m (P-YOLOv5m), and GhostNet-YOLOv5m (G-YOLOv5m) on the substation image dataset in this paper. The results are shown in Table 3.

Combining with Figure 11, we can conclude that compared to the original model, the FasterNet-YOLOv5m proposed in this paper reduces the parameter number by 44.03%, increases the FPS by 122.7%, and decreases the average precision by 2.8%. The G-YOLOv5m model exhibits high precision, low parameter number, flexibility, and near real-time performance, making it a powerful object detection model. Despite having the highest average precision (65.5%) and the lowest parameter number (10.85 M), the G-YOLOv5m model has a slower computational speed but still achieves an FPS of 207.72. This demonstrates its reasonable near real-time performance. The FasterNet-YOLOv5m model adopts the lightweighting approach of FasterNet and improves the C3 module of the original model, which reduces the parameter number and computational complexity, thereby improving the computational speed of the model. In comparison, the G-YOLOv5m model has a slower computational speed. The FasterNet-YOLOv5m model achieves a good balance between precision and speed. Although it may not match the average precision of the G-YOLOv5m model (65.5%), it still provides reasonable detection precision while offering faster computational speed. Therefore, the FasterNet-YOLOv5m model is suitable for applications which require real-time performance.

In contrast, the F-YOLOv5m model achieves the best balance among parameter number, real-time detection performance, and precision. Hence, this paper selects the lightweight model F-YOLOv5m as the foundation for further improvements.

### 4.2. Comparative Experiment of Improved Loss Function

The NWD-CIoU loss function proposed in Section 3.2 of this paper is applied to the F-YOLOv5m network architecture. Under the condition of the same input feature map size, comparative experiments were conducted with different fusion weights, and the results are shown in Table 4.

Based on Figure 12, it can be concluded that when CIoU accounts for 95% of the loss function and NWD accounts for 5%, the model achieves an average precision of 65.6%, which is 1.1% higher than that of F-YOLOv5m. Additionally, the convergence speed of the total loss remains almost the same.

To verify the regression improvement of the proposed NWD-CIoU function in this paper, experiments were conducted to compare the localization and overall losses using the CIOU, Focal-CIoU [25], and the NWD-CIoU function with the best fusion weight of 0.95. The results are shown in Figure 13. 

Table 5 presents the performance metrics of the model obtained in the last training epoch after 200 iterations. Combined with Figure 13, it can be observed that compared to the default complete intersection over union (CIoU) loss function used in the original YOLOv5 training, Focal-CIoU reduces the localization loss by 22.65% and the overall loss by only 13.86%. However, it results in a slight decrease in average precision by 1.4%. On the other hand, the proposed NWD-CIoU function in this paper outperforms both of the aforementioned loss functions. Compared to the default CIoU loss function used in the original YOLOv5, the NWD-CIoU function reduces the localization loss by 35.37% and the overall loss by 19.74%. Moreover, it demonstrates a clear advantage in terms of convergence speed.

The performance indicators of the best-performing model in 200 rounds of training, as shown in Table 6, using the original YOLOv5 with the fully intersection over union (IoU) loss function, yielded an average precision of 62.6%. After replacing the fully IoU with Focal-CIoU, there was a slight decrease in average precision. However, with our improved loss function NWD-CIoU, the average precision increased by 3% compared to the original YOLOv5. Combining the total loss value curves shown in Figure 13, it is visually evident that the loss value of NWD-CIoU decreases faster among the three loss functions and ultimately reaches a lower converged value. In conclusion, our improved NWD-CIoU loss function not only enhances the accuracy of the original algorithm but also significantly accelerates the convergence speed of the model.

### 4.3. Comparative Experiment of the Dynamic Head

The performance of the model was compared by altering the stack number (block) of the dynamic head in the detection head, using the improved model with a fusion weight of 0.95 (F-YOLOv5m-95) for the loss function in Section 3.2 as the baseline network. The experimental results are shown in Table 7.

Based on Figure 14, it can be concluded that the F-YOLOv5m-95, which is a lightweight and loss-function-improved version of the base model YOLOv5m, achieves a reduction in parameter number and an increase in FPS. However, it suffers from a decrease in average precision. Introducing the dynamic head module effectively solves this problem. When block = 2, which means two dynamic head modules are stacked, the model achieves an average precision of 67.6%, which is a 0.3% improvement over YOLOv5m. The FPS reaches 196.86, which is a 43.5% increase compared to YOLOv5m, and the parameter number is reduced to 12.36M, which is a 41.0% decrease compared to YOLOv5m. Overall, this model demonstrates the best performance and is referred to as YOLO-Substation-large (YOLO-SS-Large) in this paper.

### 4.4. Comparative Experiment of YOLO-SS-Large

In order to validate the effectiveness of each improvement module in the YOLO-SS-Large model, multiple rounds of ablation experiments were conducted in this study. The experimental results, as shown in Table 8, indicate the following: after the lightweight model based on the idea of FasterNet was improved, the FPS increased by 136.57 and the parameter count decreased by 44%, but there was a noticeable decrease in average precision and recall. After applying the NWD-CIoU loss function designed in this study, the total loss of the model decreased by 19.74%, resulting in improved convergence performance, and the average precision increased by 0.9%. Additionally, introducing the dynamic head module further increased the average precision by 2% while maintaining a stable FPS of 196.86. The improved model (YOLO-SS-Large) achieved an average precision of 67.6% while reducing the parameter count of the original model by 41.0%, resulting in a 0.3% improvement compared to the base model. Finally, by combining with Figure 15, the proposed model achieved an average precision of 69.1%, a 24.3% decrease in total loss and a 43.5% increase in FPS compared to the original model with a 27.8% increase in parameter count. The overall performance of the model is significantly better than YOLOv5m and YOLOv5l. The results of the ablation experiments demonstrate the effectiveness of each improvement module in the YOLO-SS-Large model.

To further validate the performance of the YOLO-SS-Large model in detecting defects in the collected substation dataset in this study, a comparison was made between the proposed YOLO-SS-Large model and other state-of-the-art object detection methods. These methods include the two-stage object detection model Faster R-CNN in the RCNN family, the single-stage object detection model SSD with SSD-MobileNetv2, and the YOLO series models YOLOv7-tiny and YOLOv8n.

The experimental results, as shown in Table 9, led to the following conclusions: the proposed lightweight model, YOLO-SS-Large, achieves a balance between parameter count, average precision, and FPS, outperforming the other models. It only has a slight increase in parameters (2.19M) compared to YOLOv8n, while improving the average precision by 2.3%. Furthermore, the total loss of this model after 200 training epochs is significantly lower than other models, including YOLOv7-tiny, while achieving a 6.5% higher average precision than YOLOv7-tiny. Therefore, the proposed YOLO-SS-Large model exhibits superior comprehensive detection performance in substation defect detection compared to other similar methods.

## 5. Conclusions

This paper addresses the issues of high computational requirements, large parameter numbers, and compromised average precision in the traditional YOLO algorithm for defect detection in substations. To mitigate these problems, a lightweight and high-performance improved model called YOLO-SS-Large is proposed.

The YOLO-SS-Large model is based on the YOLOv5m as the basic network, incorporating the lightweighting concept of FasterNet to construct the F-YOLOv5m model. Comparative experiments are conducted with three other improved models: MobileNet-YOLOv5m, PP-LCNet-YOLOv5, and GhostNet-YOLOv5. It was observed that F-YOLOv5m achieves the best balance between average precision and parameter number. By comparing experiments with different fusion weight ratios, it was concluded that a fusion weight of 0.95 leads to the maximum performance improvement. Through comparative experiments on the depth of the dynamic head module, it was concluded that the model proposed in this paper performs relatively best when block = 2.

The YOLO-Substation-Large (YOLO-SS-large) model proposed in this paper combines three improvements and achieves an average precision of 67.6%, which is 0.3% higher than that of YOLOv5m. The FPS reaches 196.86, showing 43.5% higher than that of YOLOv5m. The parameter number is reduced to 12.36M, which is 41.0% lower than that of YOLOv5m. The proposed model demonstrates superior detection performance in substation defect detection compared to other models in its category, including YOLOv7-tiny, YOLOv8n, and Faster R-CNN. These results demonstrate that the proposed YOLO-SS-large model significantly reduces computational requirements while achieving faster computation. The model effectively mitigates the decrease in average precision by lightweighting through the use of a novel loss function and dynamic head module. It achieves the balance between detection precision and speed, providing new insights and theoretical models for reliable and efficient intelligent detection in substations, thereby having practical engineering significance.

## Figures and Tables

**Figure 1 sensors-23-08080-f001:**
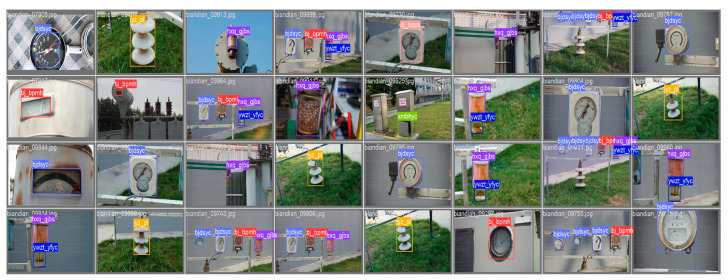
Part of substation image data.

**Figure 2 sensors-23-08080-f002:**
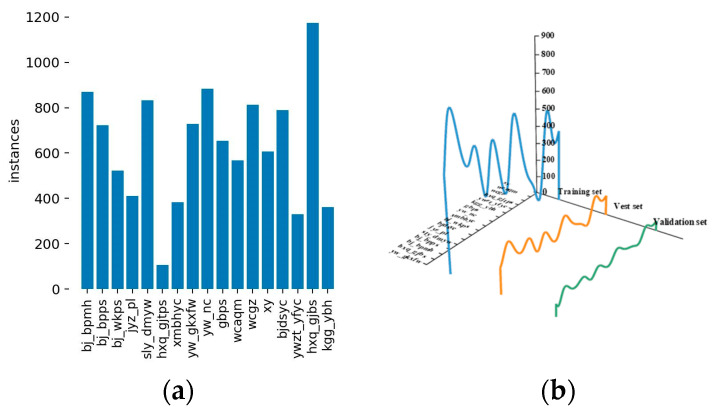
Distribution characteristics of the substation image dataset: (**a**) distribution of the training set categories; (**b**) the independent and identically distributed (i.i.d.) nature of the dataset.

**Figure 3 sensors-23-08080-f003:**
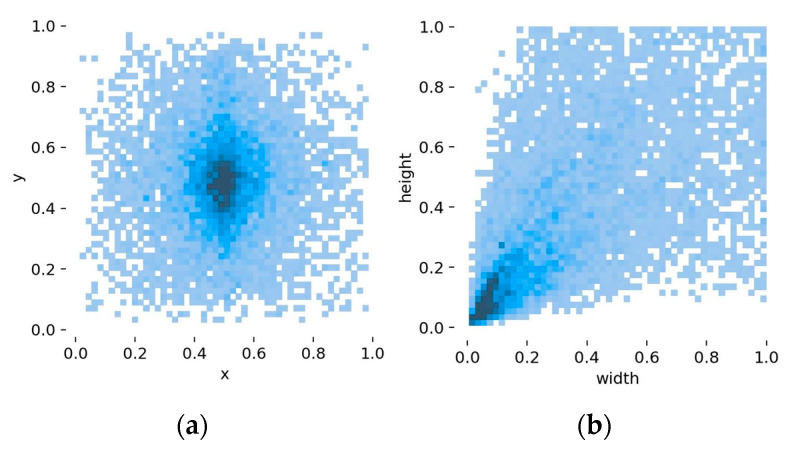
Distribution of the real bounding box: (**a**) distribution of the center point and (**b**) distribution of the length and width.

**Figure 4 sensors-23-08080-f004:**
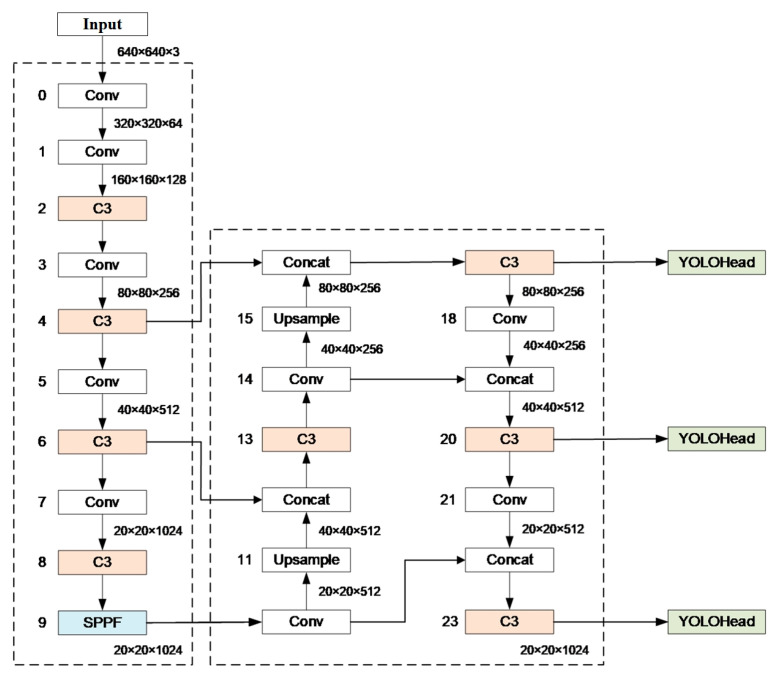
Overall structure of YOLOv5.

**Figure 5 sensors-23-08080-f005:**
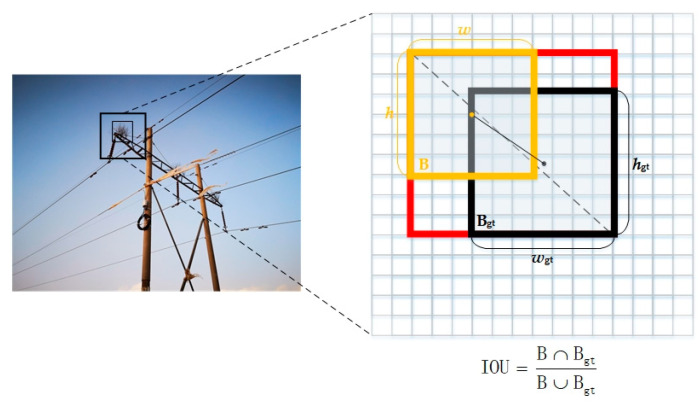
The principle of CIoU.

**Figure 6 sensors-23-08080-f006:**
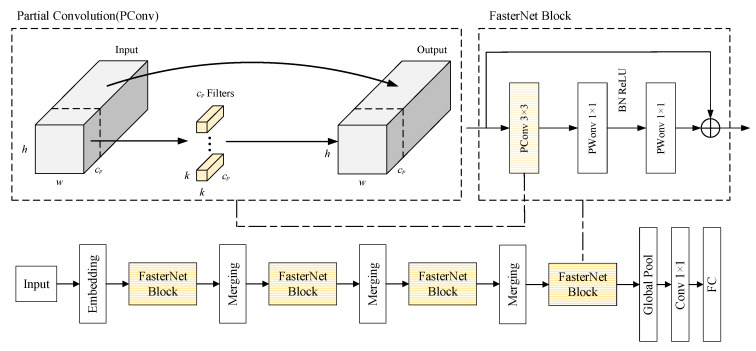
Overall FasterNet architecture.

**Figure 7 sensors-23-08080-f007:**
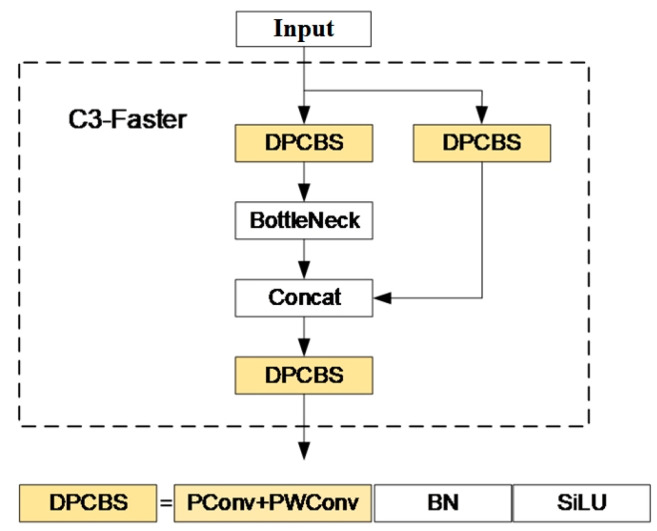
C3-Faster module structure.

**Figure 8 sensors-23-08080-f008:**
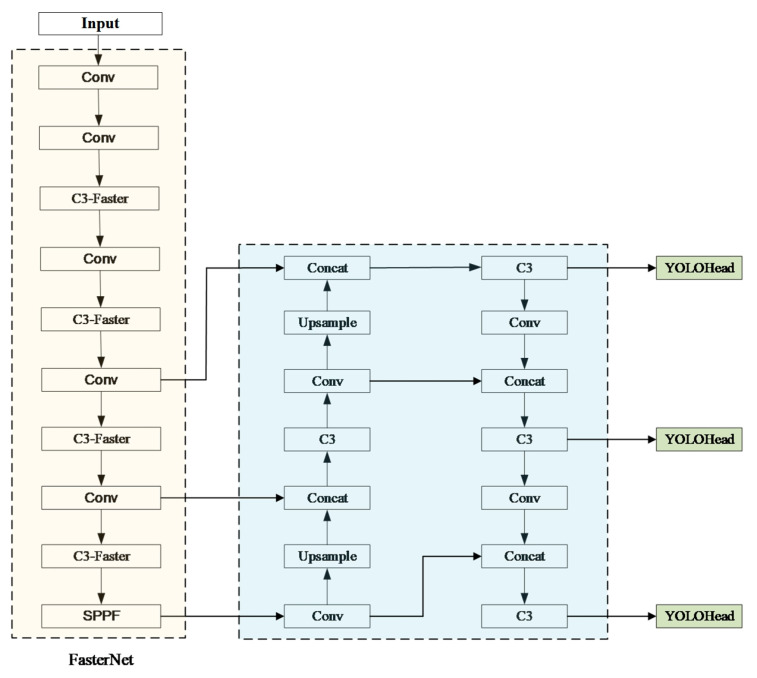
Structure of FasterNet-YOLOV5m.

**Figure 9 sensors-23-08080-f009:**
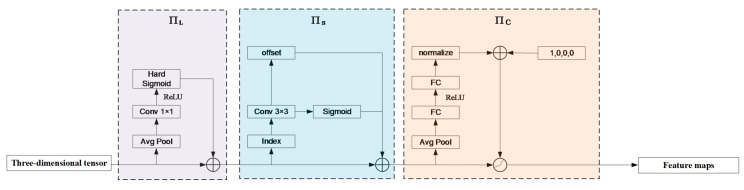
Dynamic head module.

**Figure 10 sensors-23-08080-f010:**
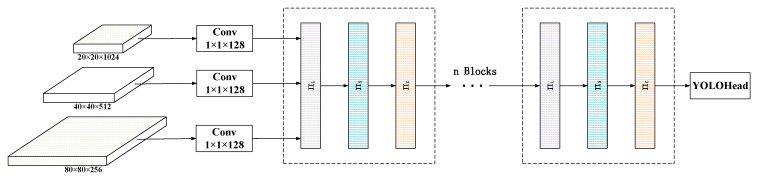
Dyhead-YOLOv5m module.

**Figure 11 sensors-23-08080-f011:**
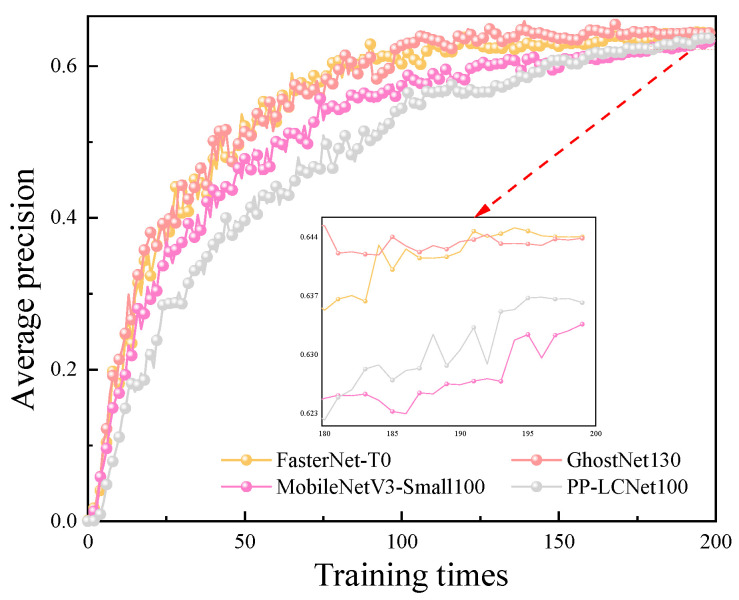
Average precision comparison of lightweight models.

**Figure 12 sensors-23-08080-f012:**
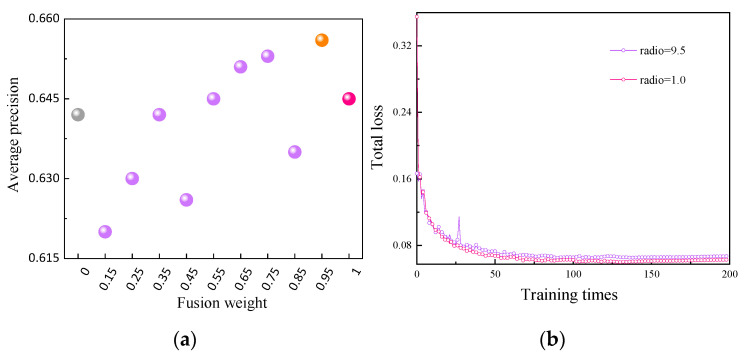
Improved results of the loss function: (**a**) average precision and (**b**) total loss.

**Figure 13 sensors-23-08080-f013:**
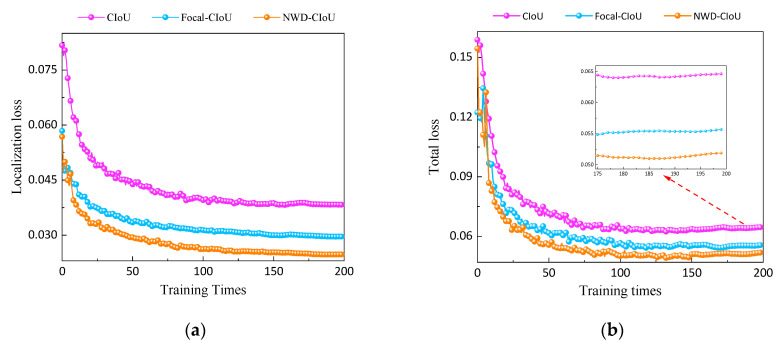
The convergence process of the loss function: (**a**) localization loss and (**b**) total loss.

**Figure 14 sensors-23-08080-f014:**
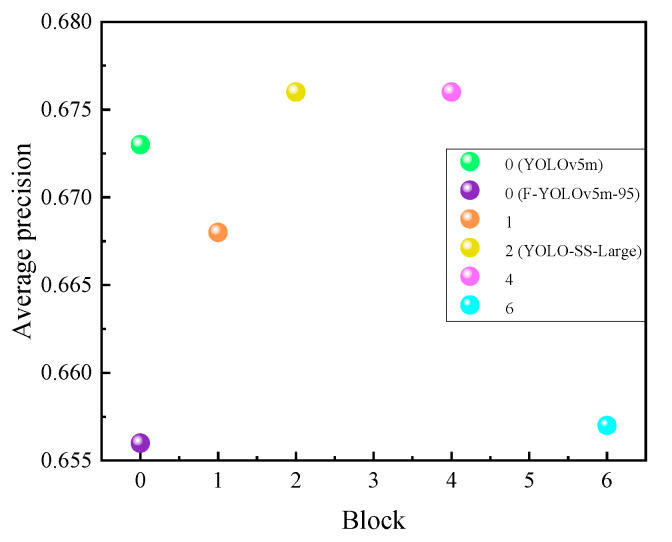
Experimental results of the improved detection head.

**Figure 15 sensors-23-08080-f015:**
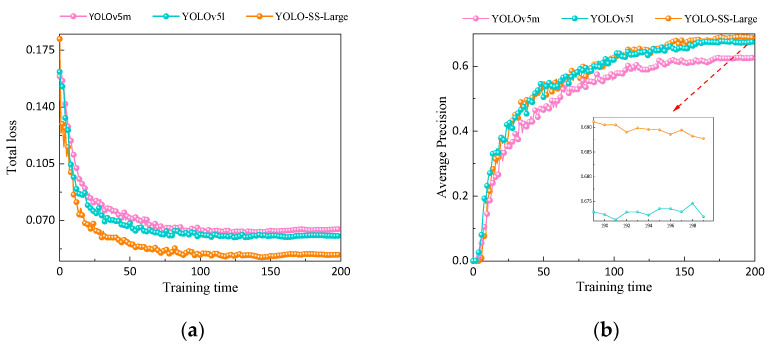
Improved results of YOLO-SS-Large: (**a**) total loss and (**b**) average precision.

**Table 1 sensors-23-08080-t001:** Dataset of substation images.

Numbering	Defect Category	Label ID	Number of Images
1	Hanging suspensions	yw_gkxfw	729
2	Discoloration of respirator silicone	hxq_gjbs	1174
3	Blurred dials	bj_bpmh	877
4	Broken dials	bj_bpps	723
5	Oil on the ground	sly_dmyw	833
6	Broken insulators	jyz_pl	410
7	Abnormal meter readings	bjdsyc	789
8	Shell damage	bj_wkps	523
9	Abnormal closure of box door	xmbhyc	383
10	Bird’s nest	yw_nc	883
11	Cover damage	gbps	654
12	Pressure plate condition	kgg_ybh	376
13	Abnormal oil level status	ywzt_yfyc	233
14	Damage to respirator silicone	hxq_gjtps	173
15	Not wearing work clothes	wcgz	787
16	No safety helmet	wcaqm	546
17	Smoking	xy	584

**Table 2 sensors-23-08080-t002:** Comparison of different versions of FasterNet models.

Network Model	Parameter Number/M	Precision/%
**FasterNet-T0**	**3.9**	**71.9**
FasterNet-T1	7.6	76.2
FasterNet-T2	15.0	78.9
FasterNet-S	31.1	81.3
FasterNet-M	53.5	83.0
FasterNet-L	93.5	83.5

**Table 3 sensors-23-08080-t003:** Experimental results of model lightweighting.

Network Model	Recall Rates	Average Precision/%	Parameter Number/M	FPS
YOLOv5m	0.653	67.3	20.94	111.29
M-YOLOv5m	0.6	62.4	10.88	286.64
P-YOLOv5m	0.593	63.6	10.94	290.42
G-YOLOv5m	0.618	65.5	10.85	207.72
**F-YOLOv5m**	**0.641**	**64.5**	**11.72**	**247.86**

**Table 4 sensors-23-08080-t004:** Comparison of fusion weights of the loss function.

Fusion Weight	Precision	Recall	Average Precision/%
1.00	0.709	0.641	64.5
**0.95**	**0.786**	**0.601**	**65.6**
0.85	0.730	0.575	63.5
0.75	0.684	0.623	65.3
0.65	0.758	0.61	65.1
0.55	0.704	0.638	64.5
0.45	0.688	0.62	62.6
0.35	0.768	0.588	64.2
0.25	0.728	0.589	63.0
0.15	0.736	0.615	62.0
0.00	0.663	0.645	64.2

**Table 5 sensors-23-08080-t005:** The convergence process of the loss function.

Loss Function	Total Loss	Localization Loss
CIoU	0.038303	0.064629
Focal-CIoU	0.029626	0.055672
**NWD-CIoU**	**0.024755**	**0.051871**

**Table 6 sensors-23-08080-t006:** Improved results of the loss function.

Loss Function	Precision	Recall	Average Precision/%
CIoU	0.701	0.6	62.6
Focal-CIoU	0.662	0.609	61.2
**NWD-CIoU**	**0.786**	**0.601**	**65.6**

**Table 7 sensors-23-08080-t007:** Experimental results of the dynamic head.

Block	Recall Rates	Average Precision/%	Parameter Number/M	FPS
0 (YOLOv5m)	0.653	67.3	20.94	111.29
0 (F-YOLOv5m-95)	0.601	65.6	11.72	247.86
1	0.627	66.8	12.35	131.57
**2** **(****YOLOv5m-SS-Large****)**	**0.637**	**67.6**	**12.36**	**196.86**
4	0.633	67.6	12.92	152.60
6	0.612	65.7	12.35	196.13

**Table 8 sensors-23-08080-t008:** Experimental results of the ablation experiment.

Model	Total Loss	Recall Rates	Average Precision/%	Parameter Number/M	FPS
Base model (YOLOv5m)	0.06463	0.653	67.3	20.94	111.29
+FasterNet	0.05187	0.641	64.5	11.72	247.86
+FasterNet+NWD-CIoU	0.04748	0.601	65.6	11.72	247.86
**+FasterNet+NWD-CIoU+Dynamic head** (YOLO-SS-Large)	**0.0489**	**0.637**	**67.6**	**12.36**	**196.86**
YOLOv5l	0.0605	0.612	65.7	12.35	146.13

**Table 9 sensors-23-08080-t009:** Horizontal comparison experiment.

Model	Total Loss	Average Precision/%	Parameter Number/M	FPS
Faster-RCNN	1.653	40.3	138.36	—
SSD-MobileNetv2	4.692	44.2	8.8	84.7
YOLOv7-tiny	0.046	61.1	6.04	259.0
YOLOv8n	2.689	65.3	10.16	273.3
**YOLO-SS-Large**	**0.049**	**67.6**	**12.35**	**196.8**

## Data Availability

The foundational data for this article is derived from a certain 110 kV substation. The derived data generated in this study will be shared with the respective authors upon reasonable request.

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
