# Peer review of "YOLO-SS-Large: A Lightweight and High-Performance Model for Defect Detection in Substations"

_sensors, 2023, doi:10.3390/s23198080_

Round 1

Reviewer 1 Report

This paper proposes YOLO-SS-large based on YOLOv5m, achieves an average precision improvement slightly, greatly improves FPS and reduces parameters, analyzes the results of the original model, lightweight model, loss function optimization, and head optimization. However, there are still the following issues that the authors need to consider and improve in the paper.

1. This article mainly focuses on the problems of untimely and slow response in the intelligent inspection process. Therefore, in the process of establishing the dataset, it is necessary to increase the proportion of images obtained by the image acquisition method of this model. In section 2.1, the methods of image acquisition, including on site photography, capturing survey footprint and collecting historical data, as well as the total number of images obtained, were introduced, but the image proportions obtained by various methods were not introduced.

2. Add parameter explanations in formulas 1-4.

3. The author created a dataset of 17 defects, but in Figure 2b, the adaptive anchor boxes jointly generated by 11 defect detection categories are represented. It is recommended that the author explain in the paper why only 11 of them are introduced.

4. In Table 1, the number of images is very uneven, and the sample size of some types of datasets is too small. However, the authors did not mention in the article whether overfitting would occur, which would affect the final conclusion. Suggest increasing the number of images with small sample sizes to increase the reliability of the results.

Reviewer 2 Report

1.Lack of a large number of comparative experiments, such as comparing the introduced NWD-CIoU loss function with more other loss functions, comparing the introduced Dynamichead with other detection heads, and comparing the improved model with other advanced lightweight object detection models, such as MobileNet, FasterNet, etc. Or compare it with more mainstream target models, such as YOLOv8, DETR, etc., to verify the reliability of the proposed model.

2.Was the dataset used in this study collected through experiments? If so, it is recommended to make it publicly available to increase research contributions. If not, it should be clearly stated and cited.

3.Figure 2 could benefit from some clarity improvements. The original loss functions, CIOU and NWD, can be used to create theoretical graphs that would enhance readability.

4.There are many missing units in the tables. Please complete them again to ensure accuracy.

5.Table 5's scheme description appears to be somewhat confusing. The labeling for the 'block' column needs to be clarified.

6.To improve readability, you can highlight the improvement of indicators in bold in each comparison table.

7.It seems that the full text lacks a comparison chart of the test results, which should be included in the comparative experiment for better understanding.

This article's English writing quality can still be improved. It is recommended to polish and make appropriate modifications to make it more fluent and natural.

Reviewer 3 Report

1. English needs to be improved.

2. The dataset can be better described. Since authors introduced the substation dataset first, more details could be given on the dataset since it could also be contribution to the paper. This substation dataset should have some obvious differences with online open dataset.  So the introduction to this dataset can be strengthened, including the training process.

3. Since the proposed method is mostly based on YOLOv5, it is better that an updated version of Yolo be compared other than just YOLOv5m. Newer methods are available.

4. There is relatively little discussion on the core theory of how to reduce the number of parameters used, and at least algorithm comparisons in this area can be increased, such as EfficientNet, newer version of YOLO and MobileNet. The implementation detail should also be provided in the paper .

5. The characteristics of the dataset images of the power plant can be introduced, which can help explain why Yolo is used instead of others. And  the relationship between image recognition of power grid can be introduced.

6.The comparison and discussion with G-YOLOv5m can be more in-depth. It is better if ablation experiments can be conducted.

7. There are too many "In summary" and "In conclusion". The paper structure could be improved.

8. For table 3-5:units for Parameter number should be given .

9. Figure 11 and 10 are almost the same with no more information be given.

10. The figures quality could be improved.

Reviewer 4 Report

1. The English of the manuscript needs to be re-edited, for example:

(1)the initials should be ..

(2)there is a typo in Line 334.

(3)the font of the article is inconsistent.

2. For the literature discussed in the introduction, please create a table to compare with the proposed method.

3. The images in Figures 1 and 2 are poor.

4. In some figures, the X-axis and Y-axis have no units.

5. Input signals need to be added to Figures 2, 5, and 6.

6. Figure 7 requires adding input and output signals.

7. Balls of different colors in Figure 11 need to be marked with their meanings in the picture.

The English of the manuscript needs to be re-edited, for example:

(1)the initials should be ..

(2)there is a typo in Line 334.

(3)the font of the article is inconsistent.

Round 2

Reviewer 2 Report

1. Please ensure that recall rates and precision are expressed as percentages throughout the text and tables. Kindly review the entire document to ensure consistency.

2. Please pay attention to the formatting of the tables throughout the full text to ensure consistency.

The English writing quality of this article has improved to some extent, but there is still room for improvement.

Reviewer 3 Report

The paper aims to identify the substation deects of power grid, and thus propose an called YOLO-SS-Large, based on YOLOv5m.The reported performance is good in precision and lightweight.

The text format can be improved. 

Reviewer 4 Report

No issues.